# Joint Unsupervised Domain Adaptation and Semi-Supervised Learning for Multi-Sequence MR Abdominal Organ Segmentation

Jiahui He[1,2][0009−0006−6847−2639], Lei Wu[2,3][0009−0007−2719−1416], Wenbin Liu[1][0000−0001−9091−3177], Zaiyi Liu[2,3][0000−0002−9307−8522], and Gang Fang[1][0000−0001−9847−114X]

[1] Institute of Computing Science and Technology, Guangzhou University, Guangzhou, 510006, China
[2] Department of Radiology, Guangdong Provincial People's Hospital (Guangdong Academy of Medical Sciences), Southern Medical University, Guangzhou 510080, China
[3] Guangdong Provincial Key Laboratory of Artificial Intelligence in Medical Image Analysis and Application, Guangzhou 510080, China
wulei@gdph.org.cn, liuzaiyi@gdph.org.cn, gangf@gzhu.edu.cn

**Abstract.** Abdominal multi-organ segmentation in multiple MR sequences plays a crucial role in assisting clinical decision making. However, the annotation of multi-sequence MR images is time-consuming and labor-intensive, and the limited availability of labels constrains the advancement of multi-sequence MR abdominal segmentation. In this study, we propose a three-stage method for multi-sequence MR abdominal organ segmentation, incorporating unsupervised domain adaptation, semi-supervised learning, registration, and anatomical structure constraints. Specifically, in the first stage, pseudo-labels for T1-weighted (T1W) sequences are generated using labeled CT samples and unlabeled T1W sequence samples through unsupervised domain adaptation and semi-supervised learning. The second stage involves using a T1W-to-Multi-sequence label transfer module to share the pseudo-labels from T1W sequence with other sequences that have minimal positional differences, followed by training a multi-sequence segmentation network via semi-supervised learning. In the third stage, iterative training is conducted using the pseudo-labels generated by the segmentation model, with an Anatomy-aware module employed to enhance the accuracy of the pseudo labels. Our method achieved an average score of 81.60% and 89.83% for the organ DSC and NSD on the validation set and the average running time and area under GPU memory-time curve were 11.80s and 26888MB, respectively. Our code is available at https://github.com/Ho-Garfield/FLARE2024_he.

**Keywords:** Abdominal organ segmentation · semi-supervised learning · unsupervised domain adaptation · registration · anatomical constraints.

## 1   Introduction

Abdominal imaging, including CT and MRI, is crucial for diagnosing and assessing abdominal diseases involving organs such as the liver, kidneys, and spleen [1,2]. Accurate segmentation of abdominal organs is essential for improving disease diagnosis, lesion detection, and developing effective treatment plans [3,4].

Significant progress has been made in the segmentation of abdominal organs in CT imaging, mainly due to the high resolution of CT images and the availability of high-quality manual annotations, which have facilitated the development of effective segmentation algorithms [5]. Compared to CT, MR imaging offers a wider range of parameters, allowing for multi-parametric and multi-sequence imaging, which provides advantages in diagnosing soft tissue diseases [6]. However, the diversity of MRI sequences makes annotation more challenging, and the substantial differences between MR sequences and poor image quality have limited the exploration of multi-sequence MR-based abdominal organ segmentation. Additionally, the lack of one-to-one paired samples between CT and MRI further complicates these challenges.

To overcome these challenges, image-to-image translation-based Unsupervised Domain Adaptation (UDA) methods have been widely used. For example, CycleGAN [7], a classic UDA method, is popular in medical imaging field due to its unique structural consistency constraints and the convenience of not requiring paired data [8]. However, these one-to-one domain adaptation methods can only learn the relationship between two different domains. In the context of domain adaptation with multiple target domains, as illustrated in Figure 1, this approach requires the construction of multiple transfer networks for each sequence pair, which limits scalability. To address this issue, Yang et al. [9] proposed an unsupervised domain adaptation technique for liver segmentation by embedding images from each domain into a shared domain-invariant content space and a domain-specific style space through disentangled learning. Similarly, Xu et al. [10] employed content and style separation with generator reconstruction and style constraints, effectively transforming source domain images into multiple target domains while preserving structural consistency and minimizing domain mixing. Semi-supervised learning methods also offer promising solutions in this context [11]. For example, Chen et al [12]. leveraged cross-modal consistency between CT and MRI as a constraint and introduced a contrastive similarity loss to address cross-modal abdominal organ segmentation. Likewise, Zhao et al [13]. integrated the Mean Teacher model [14] into a UDA framework to enhance cross-modal medical image segmentation. However, these methods rarely fully leveraged the characteristic of different sequences of the same sample with highly similar structures within multi-sequence MR datasets. Instead, they address the problem of unsupervised domain adaptation for segmentation from CT to multi-sequences MR by improving deep learning models originally designed for natural images.

Unlike these approaches, our approach leverages the unique properties of medical images by introducing a T1W-to-Multi-sequence label transfer mod-

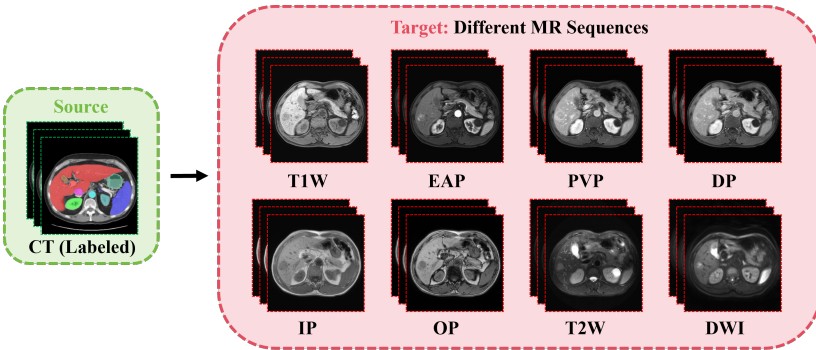

**Fig. 1.** Illustration of the task in this work. We aim to segment 13 abdominal organs in different MR sequences (e.g., dynamic contrast enhancement (including non-contrast phase (T1-weighted (T1W)), early-arterial phase (EAP), portal venous phase (PVP), delay phase (DP)), t2-weighted (T2 W), diffusion-weighted image (DWI) and in-phase (IP)/out-of-phase (OP)) by learning from existing annotated CT datasets without annotations in the target domains.

ule with filtering capabilities, facilitating label sharing across multiple MR sequences. This approach transforms the one-to-multiple domain adaptation problem into a simpler one-to-one domain adaptation problem. To further enhance segmentation accuracy, we propose an iterative training framework that refines pseudo-labels using anatomical structure constraints. Specifically, our apppoach includes three parts: 1) CT to T1W image translation and pseudolabel generation: we use CycleGAN to translate CT images into T1W images (fake T1W), and then perform semi-supervised learning to genarate T1W pseudo-labels by using fake T1W images, along with their paired CT labels and real T1W images. 2) Label transfer and multi-sequence segmentation: we employ the T1W-to-Multi-sequence label transfer module to share labels among sequences with minimal positional differences, conduct semi-supervised training with labeled CT data to develop a multi-sequence MR segmentation model, and subsequently use this model to generate pseudo-labels for the LLD-MMR(Liver Lesion Diagnosis on Multi-phase MRI) [15] and AMOS [16] datasets. 3) Anatomy-aware refinement and iterative learning: an Anatomy-aware module and iterative learning strategy were introduced to enhance the model's performance and robustness. In summary, our main contributions are threefold:

- We propose a multi-sequence MR abdominal organ segmentation method based on unsupervised domain adaptation and semi-supervised learning.
- We introduce a T1W-to-Multi-sequence label transfer module to facilitate label sharing across multiple sequences and improve pseudo-label accuracy, along with an Anatomy-aware module to enhance these processes.
- Our method achieves strong performance on the multi-sequence MR abdominal multi-organ datasets.

## 2   Method

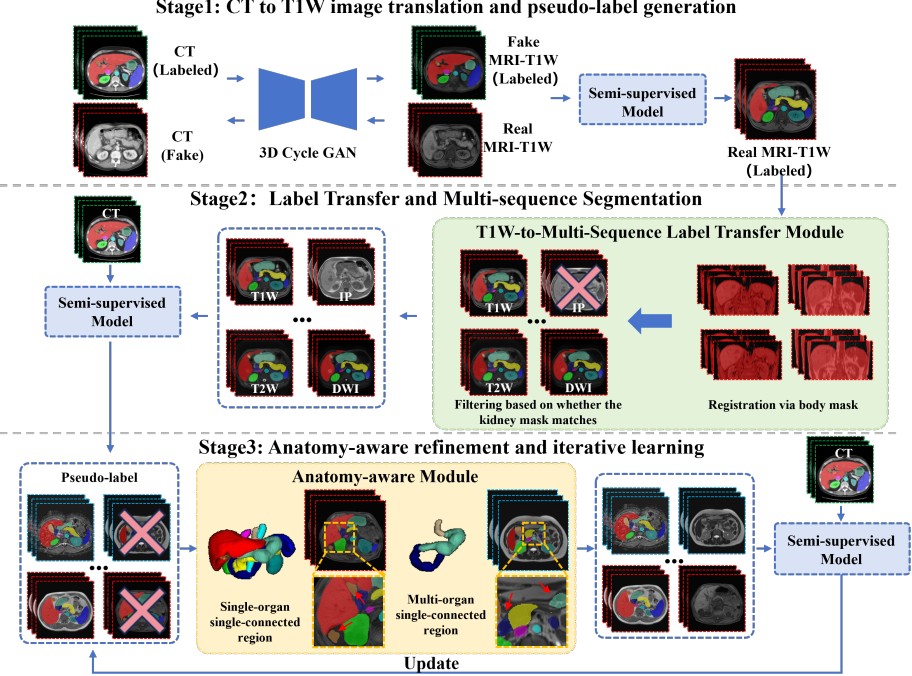

**Fig. 2.** Flowchart of our method, where green dashed images are from the CT dataset, red dashed images are from the LLD-MRI dataset, blue dashed images are from the AMOS dataset, and yellow dashed boxes in Stage 3 highlight parts that do not meet anatomical constraints (e.g., non-unique organ components or disconnected duodenum and stomach). In Stage 1, 498 CT samples are used for CycleGAN training. In Stages 2 and 3, 100 manually annotated CT samples and 400 pseudo-labeled CT samples (slice > 150) are used.

As shown in Figure 2, we propose a three-stage method for abdominal organ segmentation in multiple MR sequences. The first stage involves generating pseudo-labels for the T1W sequence, the second stage focuses on training a multi-sequence MR segmentation model, and the third stage is dedicated to iterative training the segmentation model on multiple datasets. The specific details of each stage are described below.

### 2.1   Preprocessing

Since our method includes both domain transfer and semantic segmentation models, our preprocessing techniques differ for these tasks. Specifically, for both

models, we first performed resampling, patient position readjustment and adjusting grayscale range for initialization. Particularly, for the style transfer model (CycleGAN) [7], we also used translation registration to align the anatomical structures of the T1W sequence and CT.

- **Resampling.** We use B-spline interpolation to resample the images and nearest-neighbor interpolation for the labels, adjusting the pixel spacing to $1.2 \times 1.2 \times 3$ to reduce GPU memory usage.
- **Patient position readjustment.** We first normalize all positions to the LPS orientation. However, for the LLD-MMR dataset, we observed that some samples have different Z-axis orientations. To unify the body orientation, we used Otsu's method [17] to obtain the body mask of the in-phase sequence samples and decided whether to flip the samples along the z-axis based on the size of the black regions within the body mask on either side of the z-axis, ensuring consistent z-axis orientation for the LLD-MMR dataset.
- **Adjust grayscale range.** To facilitate better transfer and semantic segmentation, we first converted the CT and MR sequences to a grayscale range. For CT images, the window level is first set to 40 and the window width to 400, followed by scaling the intensity to a range of 0-255. For MR images, the intensity is directly scaled to 0-255.
- **Translation registration.** To ensure that the parts of the input to the network correspond better during cropping for CycleGAN training, we first performed translation registration on the CT data based on the existing labels, with the registration target being "FLARE22_Tr_0001" (the first in alphabetical order), to remove non-abdominal regions. Then, we used Otsu's method to obtain body masks for all samples in the CT and LLD-MMR datasets, and performed translation registration on the T1W modality of the LLD-MMR dataset using the body mask, with the registration target being "MR745_6_C-pre" (the first in alphabetical order). To better align the datasets, we next performed translation registration on the CT and MRI datasets based on their body masks.

### 2.2   Semi-supervised Model

As shown in Figure 2, we use an identical semi-supervised model structure for Stages 1 to 3. We adopt the Mean Teacher model [14], which includes both a student model and a teacher model, where the student model is updated using standard backpropagation, and the teacher model is updated as an exponential moving average (EMA) of the student model's weights. The student and teacher models use similar network structures, where the base network architecture is a 3D U-Net(Figure 3). The model leverages unlabeled data by imposing a consistency constraint between the outputs of the student model and the outputs of the teacher model when given perturbed versions of the input data, thereby effectively utilizing the unlabeled data to enhance segmentation performance.

Specifically, for unlabeled data (e.g., with a shape of (X, Y, Z)), we utilize the consistency constraint between the outputs of the teacher model and the

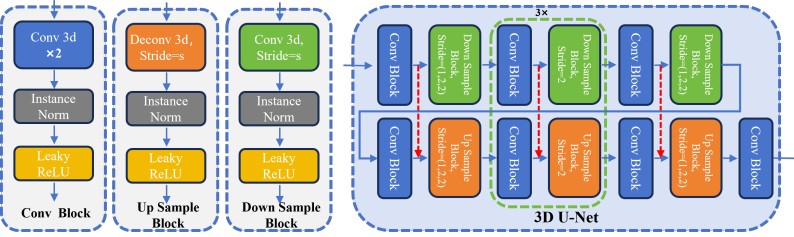

**Fig. 3.** Structure and Layer Configuration of the 3D U-Net Model Utilized in the Study

student model to regularize the model. The Mean Squared Error (MSE) used as the consistency loss is defined as follows, where $K$ represents t-he number of classes, and $p_{i,k}^{(t)}$ and $p_{i,k}^{(s)}$ correspond to the outputs of the teacher and student models, respectively.

$$L_{con} = -\frac{1}{Z \times Y \times X \times K} \sum_{i}^{Z \times Y \times X} \sum_{k=0}^{K} l_{i,k}(p_{i,k}^{(t)} - p_{i,k}^{(s)})^2 \tag{1}$$

Regarding labeled data, we perform data augmentation and employ deep supervision [18], with the supervised loss defined as a combination of multi-class Dice loss and multi-class cross-entropy loss [19]. In practice, $\alpha_{\{1,2,3\}}$ were halved with each decrease in resolution, leading to $\alpha_{i+1} = \frac{a_i}{2}$. All weight factors were ultimately normalized to sum to 1.

$$L_{deep} = \alpha_0 loss_{sup}^{Z \times Y \times X} + \alpha_1 loss_{sup}^{Z \times \frac{Y}{2} \times \frac{X}{2}} + \alpha_2 loss_{sup}^{\frac{Z}{2} \times \frac{Y}{4} \times \frac{X}{4}} + \alpha_3 loss_{sup}^{\frac{Z}{4} \times \frac{Y}{8} \times \frac{X}{8}} \tag{2}$$

Therefore, the objective loss function of our semi-supervised model is defined as Equation 3,where $\lambda$ is a dynamic parameter that gradually increases according to $\lambda = 0.1 \times \exp(-5.0 \times (1.0 - percentage_{iter})^2)$ ,where $percentage_{iter}$ represents the percentage of the current iteration relative to the total number of iterations.

$$TotalLoss = \lambda L_{con} + L_{deep} \tag{3}$$

### 2.3  CT to T1W image translation and pseudo-label generation

In this stage, we first perform style transfer from CT to the T1W and then use transferred data for semi-supervised learning. Specifically, we use the CT labels (including manually annotated labels and pseudo-labels generated by the FLARE22 winning algorithm [20]) as the labels for the transferred pseudo-T1W sequence and conduct semi-supervised learning with the real T1W sequence to obtain pseudo-labels for the real T1W sequence. The specific details are as follows:

**CT to T1W Sequence Style Transfer.** As shown in Figure  2, style transfer is conducted exclusively from CT to the T1W. We selected 100 manually annotated CT samples and 398 CT pseudo-labeled samples (sorted by name) as the source domain for style transfer (Note that the 498 CT samples, which have been processed through Translation registration, were used exclusively in the first stage.), while 498 unlabeled T1W sequence MR samples from the LLD-MMR dataset (with names containing "pre") were used as the target domain for style transfer. We chose 3D CycleGAN as the transfer model from CT to T1W sequence because CycleGAN uses L1 loss as the cycle consistency loss (as shown in Equation 4, 5, where G is the generator from CT to T1W sequenceand F is the generator from T1W sequence to CT). This loss effectively maintains anatomical consistency, allowing the CT labels to be used as pseudo-labels for the generated fake T1W sequence.

$$Loss_{cycle_{CT}} = E_{x_{CT}}[||F(G(x_{CT})) - x_{CT}||_1] \tag{4}$$

$$Loss_{cycle_{T1W}} = E_{y_{T1W}}[||G(F(y_{T1W})) - y_{T1W}||_1] \tag{5}$$

**Semi-Supervised Training with Real and Fake T1W Sequences.** To obtain a network for segmenting the T1W sequence, we perform semi-supervised training using both real T1W sequence data from the LLD-MMR dataset and the labeled fake T1W sequence data generated from CT. This model will be used to predict the T1W sequences in the LLD-MMR dataset.

### 2.4   Label transfer and multi-sequence segmentation

In the second stage, our goal is to develop a multi-sequence MR segmentation model using the T1W pseudo-labels obtained in the first stage. To achieve this, we introduce a T1W-to-Multi-sequence label transfer module, which shares labels among multiple sequences that have minimal positional differences (filtered based on pseudo-labels of the kidneys). For sequences with larger positional differences, we leverage semi-supervised learning using unlabeled data after registration. The detailed about the module is explained below.

**T1W-to-Multi-sequence label transfer module.** Given that we only have labels for the T1W sequence and not for other sequences, we need to generate labels for the same samples in other sequences. Our main idea is to first perform registration, then validate its success using pseudo-labels of the kidneys, and finally use T1W labels as the labels for the successfully registered MR sequences. This is achieved via the following steps:

i) Body mask generation and registration. Firstly, for sequences with clear body contours, such as the early-arterial phase (EAP), portal venous phase (PVP), delay phase (DP), and in-phase (IP), we use Otsu's method to create boby mask. Secondly, those masks are then used for rigid registration to align them with the T1W mask and applied the transformation matrix to the

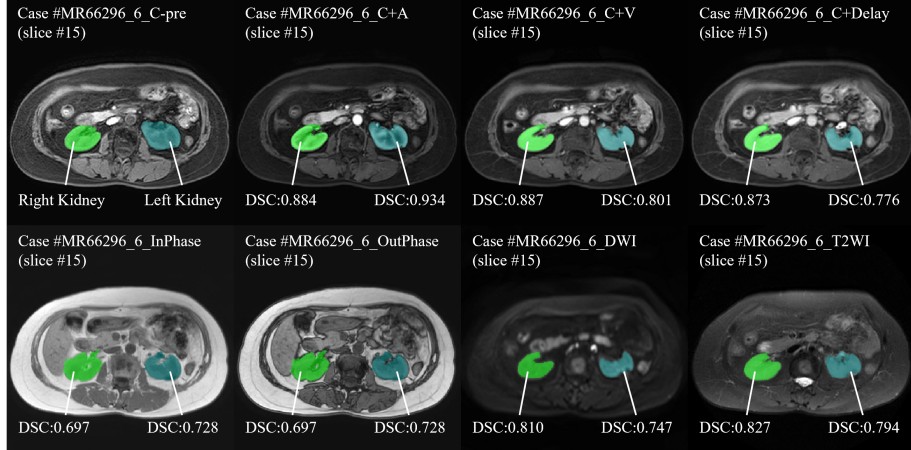

**Fig. 4.** Example of filtering other sequences based on left and right kidney pseudo-Label Dice Similarity Coefficient (DSC) scores. (In this example, IP, OP, DWI, and T2W are excluded due to the absence of DSC > 0.85 in the kidney labels)

original images. Thirdly, For the out-of-phase (OP) sequence, we apply the same transformation matrix as used for IP. Finally, for sequences with unclear body contours and significant differences, such as T2-weighted (T2W) and diffusion-weighted images (DWI), we register them to the similarly colored EAP sequence to improve alignment.

ii) Filtering incorrectly registered modalities. Since registration results are not always accurate, we further filter out incorrectly registered modalities based on kidney pseudo-labels. First, for sequences other than T1W and OP, we expanded the T1W kidney label region and apply Otsu's method within this range to obtain coarse kidney pseudo-labels for each sequence. Second, we calculate the Dice Similarity Coefficient (DSC) between these coarse pseudo-labels and the T1W kidney pseudo-label (as shown in Figure 4). Third, for EAP, PVP, and DP sequences, where temporal and resolution differences are smaller, registration is considered successful if the DSC for at least one kidney is greater than 0.85. For DWI, IP, OP, and T2W sequences, registration is deemed successful if both kidneys have DSC greater than 0.8, and at least one kidney has a DSC greater than 0.85. Using these criteria, Sequences meeting these criteria retain their samples.

**Model training based on semi-supervised learning.** As the data in the LLD-MRI dataset only includes the abdominal region, the model may lack understanding of other body parts, leading to potential mispredictions. To improve the model's learning of features from other regions of the body and enhance its robustness, we introduce manually annotated samples and pseudo-labeled samples with over 150 slices from the CT dataset. Additionally, we employ semi-

supervised learning to fully utilize unlabeled data that were not successfully registered for training the segmentation model. The trained model will be used in the third stage to generate pseudo-labels.

## 2.5 Anatomy-aware refinement and iterative learning

Through the multi-sequence segmentation model from the second stage, we can obtain pseudo-labels for all data in the MR dataset. However, these pseudo-labels contain errors, and directly using them for training would lead to suboptimal results. To further enhance the model's performance, we propose an Anatomy-aware module to eliminate pseudo-labels with anatomical errors, thereby improving the accuracy of the labels.

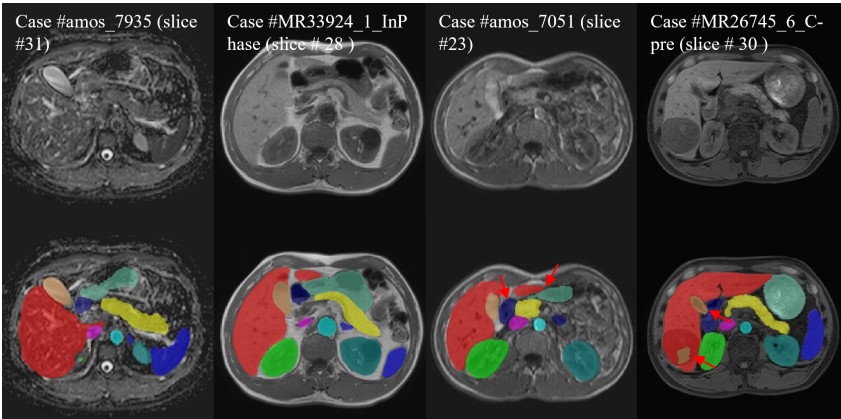

**Fig. 5.** Examples of cases meeting and not meeting anatomical structure constraints: The 1st and 2nd columns show samples that meet the constraints, the 3rd column shows a sample that does not meet the Multi-organ single-connected region constraint (the duodenum and stomach labels are not connected), and the 4th column shows a sample that does not meet the Single-organ single-connected region constraint (multiple gallbladder labels are present).

**Anatomy-aware module.** To improve the quality of the pseudo-labels, we filter the labels based on the anatomical structure of the abdomen. As shown in Figure 5. We established the corresponding anatomical constraints based on two main characteristics of normal abdominal organs: each organ has a unique connected region (Single-organ single-connected region) and the esophagus, stomach, and duodenum form a unique connected domain (Multi-organ single-connected region). Specifically, we achieve this by setting constraints 6-8. Note that the left and right adrenal glands are not subject to constraint 1. Among them, $Size_{max}$ is the size of the largest connected component of

each organ, $Size_{second}$ is the size of the second largest connected component, $Size_{esdsecond}$ is the size of the second largest connected component in the region composed of the esophagus, stomach, and duodenum, and $Size_{exist}$ and $Size_{tiny}$ are default size thresholds set to 512 and 64, respectively.

$$Size_{max} > Size_{exist} \tag{6}$$

$$Size_{second} < Size_{tiny} \tag{7}$$

$$Size_{esd_{second}} < Size_{tiny} \tag{8}$$

**Strategies to improve inference speed and reduce resource consumption.** Similar to nnU-Net [21], we use sliding window prediction for the samples. To speed up the process and reduce resource consumption, we perform predictions in half precision and set the window size to (224,160,48), with mirroring only on axes (0,2). With an initial step size of 0.5, if the total number of steps exceeds 20, we adjust the step size to (1,1,0.5) to reduce prediction time.

### 2.6   Post-processing

For the liver, stomach, gallbladder, adrenal glands, and pancreas, we remove predictions outside of this range. For the aorta and kidneys, we use the largest connected region as the label. We then establish body orientation based on the positions of the liver and kidneys, eliminating incorrect predictions in regions such as the duodenum or pancreas located above the liver and the spleen region located below the kidneys.

## 3   Experiments

### 3.1   Dataset and evaluation measures

The training dataset is curated from more than 30 medical centers under the license permission, including TCIA [22], LiTS [23], MSD [24], KiTS [25,26], autoPET [27,28], AMOS [16], LLD-MMRI [15], TotalSegmentator [29], and AbdomenCT-1K [30], and past FLARE Challenges [31,32,33]. The training set includes 2050 abdomen CT scans and over 4000 MRI scans. The validation and testing sets include 110 and 300 MRI scans, respectively, which cover various MRI sequences, such as T1, T2, DWI, and so on. The organ annotation process used ITK-SNAP [34], nnU-Net [21], MedSAM [35], and Slicer Plugins [36,37].

The evaluation metrics encompass two accuracy measures—Dice Similarity Coefficient (DSC) and Normalized Surface Dice (NSD)—alongside two efficiency measures—running time and area under the GPU memory-time curve. These metrics collectively contribute to the ranking computation. Furthermore, the running time and GPU memory consumption are considered within tolerances of 15 seconds and 4 GB, respectively.

### 3.2   Implementation details

**Environment settings** The development environments and requirements are presented in Table 1.

**Table 1.** Development environments and requirements.

| | |
|---|---|
| System | Ubuntu 22.04.2 LTS |
| CPU | Intel(R) Xeon(R) Silver 4210R CPU @ 2.40GHz |
| RAM | 8×32GB; 2400MT/s |
| GPU (number and type) | NVIDIA GeForce RTX 4090 24G |
| CUDA version | 11.8 |
| Programming language | Python 3.9.0 |
| Deep learning framework | Pytorch (Torch 2.1.0) |
| Code | https://github.com/Ho-Garfield/FLARE2024_he |

**Training protocols** To address the differences between CT and MR data domains, our approach involves two stages:

i) Style transfer using CycleGAN. We use a 3D CycleGAN network to transform CT images into T1W images. During the training phase for this style transfer, we adopt the following settings: set the batch size to 1 and randomly select samples, each sample is cropped to a voxel size of [160,160,48]. For the optimizer, we use the Adam optimizer [38] with default $\beta_1 = 0.5, \beta_2 = 0.999$. The specific configuration of CycleGAN is shown in Table 2.

ii) Training semi-supervised models. For all semi-supervised models, we maintain a consistent configuration across different models. We set the batch size to 4, with 2 samples sequentially selected from the labeled data and the remaining 2 samples randomly drawn from the unlabeled data. Each sample is cropped to a voxel size of [224,160,48], with the same data augmentation and patch sampling strategy as nnU-Net [21] employed. For the optimizer, we use SGD with momentum, where the momentum is set to 0.99, and the weight decay is set to $3 \times 10^{-5}$. The specific configuration is shown in Table 3.

## 4   Results and discussion

The results of public validation are calculated based on 110 open validation cases with ground truth. Note that the validation metrics include the standard deviation (score ± std), while the testing metrics do not, as the standard deviation is not available.

### 4.1   Quantitative results on validation set

antitative ablation experiments to evaluate the impact of introducing unlabeled MR datasets and the different methods of doing so. The results are shown in

**Table 2.** Training protocols for CycleGAN.

| | |
|---|---|
| Network initialization | Normal Initialization |
| Batch size | 1 |
| Patch size | 160×160×48 |
| Total epochs | 400 |
| Optimizer | Adam (with default $\beta_1 = 0.5, \beta_2 = 0.999$) |
| Initial learning rate (lr) | 0.0002 |
| Lr decay schedule | 1- max(0, epoch + 2 - 200 )/201 |
| Training time | 80 hours |
| Loss function | Cycle-consistency loss + GAN loss |
| Number of model parameters | 90.32M |
| Number of flops | 2312.29G |
| $CO_2$eq | 22.90 kg |

**Table 3.** Training protocols for the Semi-supervised model.

| | |
|---|---|
| Network initialization | "He" Initialization |
| Batch size | 4 |
| Patch size | 224×160×48 |
| Total iterations | 150000 |
| Optimizer | SGD with nesterov momentum ($\mu = 0.99$) |
| Initial learning rate (lr) | 0.01 |
| Lr decay schedule | Poly learning rate policy: $(1 - iterations/150000)^{0.9}$ |
| Training time | 24.24 hours |
| Number of model parameters | 33.89M |
| Number of flops | 693.53G |
| $CO_2$eq | 6.33 Kg |

**Table 4.** Segmentation Metric Results on the Validation Set

| Target | Validation | |
|---|---|---|
| | DSC(%) | NSD(%) |
| Liver | 96.56±1.12 | 97.80±1.81 |
| Right kidney | 93.82±1.92 | 92.93±3.63 |
| Spleen | 95.01±9.30 | 97.58±9.72 |
| Pancreas | 83.02±8.52 | 95.85±5.63 |
| Aorta | 88.79±7.33 | 94.27±8.77 |
| Inferior vena cava | 83.73±5.97 | 87.68±7.41 |
| Right adrenal gland | 60.23±14.15 | 78.72±15.23 |
| Left adrenal gland | 67.53±17.44 | 83.69±19.98 |
| Gallbladder | 79.56±23.21 | 79.00±24.64 |
| Esophagus | 64.07±10.98 | 83.93±12.77 |
| Stomach | 87.16±9.79 | 91.72±10.92 |
| Duodenum | 67.75±10.71 | 91.25±7.60 |
| Left kidney | 93.61±2.13 | 93.39±2.86 |
| Average | 81.60±4.66 | 89.83±4.77 |

**Table 5.** Comparison of Baseline, Semi-supervised, and Proposed Methods, where **Baseline** refers to supervised learning using only CT labels (100 manually annotated and 2000 pseudo-labels generated by the FLARE22 winning algorithm [20]). **Semi-supervised** refers to semi-supervised training using CT dataset and all unlabeled MR datasets (AMOS [16] and LLD-MMR [15]datasets).

| Target | Baseline | | Semi-supervised | | Proposed | |
|---|---|---|---|---|---|---|
| | DSC(%) | NSD(%) | DSC(%) | NSD(%) | DSC(%) | NSD(%) |
| Liver | 91.20 | 90.91 | 92.58 | 92.26 | 96.56 | 97.80 |
| Right kidney | 89.16 | 88.65 | 87.98 | 88.10 | 93.82 | 92.93 |
| Spleen | 85.85 | 86.58 | 85.50 | 86.53 | 95.01 | 97.58 |
| Pancreas | 75.82 | 87.08 | 74.11 | 85.12 | 83.02 | 95.85 |
| Aorta | 82.45 | 85.75 | 84.09 | 87.57 | 88.79 | 94.27 |
| Inferior vena cava | 67.25 | 67.63 | 70.01 | 70.90 | 83.73 | 87.68 |
| Right adrenal gland | 52.93 | 69.51 | 56.76 | 72.96 | 60.23 | 78.72 |
| Left adrenal gland | 64.18 | 78.18 | 64.50 | 78.62 | 67.53 | 83.69 |
| Gallbladder | 64.86 | 62.29 | 65.98 | 62.88 | 79.56 | 79.00 |
| Esophagus | 54.79 | 66.94 | 54.72 | 67.37 | 64.07 | 83.93 |
| Stomach | 68.20 | 71.08 | 68.00 | 70.93 | 87.16 | 91.72 |
| Duodenum | 59.32 | 79.20 | 57.94 | 79.20 | 67.75 | 91.25 |
| Left kidney | 89.79 | 90.82 | 88.71 | 89.88 | 93.61 | 93.39 |
| Average | 72.75 | 78.82 | 73.14 | 79.41 | 81.60 | 89.83 |

**Table 6.** Overview of Ablation Experiment Results. Note: **Semi(w/o CT)** means using only MRI pseudo-labels for semi-supervised learning.**Semi** refers to using T1W pseudo-labels and CT labels for semi-supervised learning. **w/o AMOS** denotes the absence of the AMOS dataset [16]. **T+Semi** involves using the T1W-to-Multi-sequence label transfer module for multi-sequence labels in semi-supervised learning. **Supervised** refers to using pseudo-labels from Stage 2 for supervised learning. **A+Semi** means filtering pseudo-labels with the Anatomy-aware module before semi-supervised learning. **Proposed** represents the optimal results from multiple iterations of semi-supervised learning with the Anatomy-aware module. All training sets include the CT dataset (manually annotated samples and pseudo-labels [20] for slices > 150) and the LLD-MRI dataset [15].

| | Stage 2 | | | | | | | | Stage 3 | | | | | |
| Target | Semi(w/o CT) | | Semi | | Semi(w/o AMOS) | | T+Semi(w/o AMOS) | | Supervised | | A+Semi(iter=1) | | Proposed | |
| | DSC(%) | NSD(%) | DSC(%) | NSD(%) | DSC(%) | NSD(%) | DSC(%) | NSD(%) | DSC(%) | NSD(%) | DSC(%) | NSD(%) | DSC(%) | NSD(%) |
|---|---|---|---|---|---|---|---|---|---|---|---|---|---|---|
| Liver | 95.46 | 95.85 | 95.64 | 96.49 | 95.64 | 96.31 | 96.03 | 97.23 | 96.24 | 97.36 | 96.38 | 97.67 | 96.56 | 97.80 |
| Right kidney | 90.89 | 88.99 | 93.21 | 93.52 | 93.57 | 93.98 | 93.35 | 91.89 | 93.73 | 92.62 | 93.79 | 92.75 | 93.82 | 92.93 |
| Spleen | 92.46 | 93.69 | 92.86 | 95.17 | 92.80 | 94.78 | 95.76 | 98.08 | 94.92 | 97.34 | 95.05 | 97.60 | 95.01 | 97.58 |
| Pancreas | 79.13 | 92.64 | 80.80 | 92.73 | 80.80 | 93.03 | 81.91 | 94.67 | 81.94 | 94.57 | 82.80 | 95.19 | 83.02 | 95.85 |
| Aorta | 84.39 | 88.11 | 89.39 | 93.15 | 89.17 | 92.87 | 87.73 | 92.91 | 87.96 | 93.51 | 88.04 | 93.29 | 88.79 | 94.27 |
| Inferior vena cava | 77.65 | 77.91 | 77.90 | 79.77 | 77.67 | 79.40 | 81.42 | 84.38 | 82.18 | 85.66 | 82.87 | 86.47 | 83.73 | 87.68 |
| Right adrenal gland | 53.42 | 70.79 | 59.11 | 77.78 | 59.63 | 77.98 | 58.52 | 76.05 | 57.70 | 75.55 | 60.16 | 78.89 | 60.23 | 78.72 |
| Left adrenal gland | 61.06 | 76.85 | 67.67 | 82.88 | 67.10 | 82.30 | 66.76 | 82.78 | 66.84 | 82.99 | 67.81 | 83.61 | 67.53 | 83.69 |
| Gallbladder | 72.60 | 68.33 | 72.35 | 71.44 | 73.02 | 72.19 | 76.04 | 74.93 | 75.97 | 75.09 | 76.64 | 75.45 | 79.56 | 79.00 |
| Esophagus | 61.70 | 78.85 | 65.80 | 83.51 | 65.16 | 83.16 | 63.86 | 82.38 | 63.90 | 83.22 | 65.26 | 84.20 | 64.07 | 83.93 |
| Stomach | 83.58 | 88.61 | 83.10 | 87.04 | 83.17 | 87.22 | 86.25 | 90.91 | 86.57 | 91.09 | 86.53 | 90.83 | 87.16 | 91.72 |
| Duodenum | 63.28 | 87.61 | 65.05 | 87.24 | 64.91 | 87.84 | 66.34 | 89.11 | 66.91 | 88.53 | 67.57 | 89.38 | 67.75 | 91.25 |
| Left kidney | 92.10 | 90.61 | 91.59 | 92.72 | 91.60 | 92.51 | 93.13 | 92.45 | 93.51 | 93.18 | 92.49 | 92.37 | 93.61 | 93.39 |
| Average | 77.52 | 84.53 | 79.57 | 87.19 | 79.55 | 87.20 | 80.55 | 88.29 | 80.64 | 88.52 | 81.18 | 89.05 | 81.60 | 89.83 |

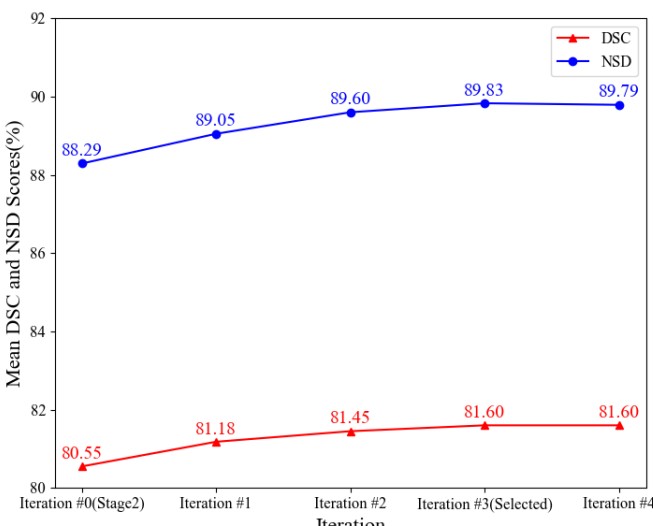

**Fig. 6.** The Impact of the Number of Iterations on DSC and NSD

Table  5. We found that when the model was trained solely using CT labels and the original CT images (window level: 40, window width: 400, scaled to the grayscale range of (0, 255)) without MR datasets, it exhibited some ability to segment MR images. However, the results were markedly suboptimal. When semi-supervised learning was applied directly to the unlabeled MR data, the performance improvement was limited due to the significant differences between CT and MR data. In contrast, compared to the direct semi-supervised learning approach, our method improved DSC and NSD by 8.46% and 10.42%, respectively, demonstrating that our method can more effectively leverage unlabeled MR data.

Secondly, to evaluate the use of unlabeled data in Stage 2 and Stage 3, as well as the effectiveness of our proposed T1W-to-Multi-sequence label transfer module and Anatomy-aware module, we conducted further quantitative ablation experiments. As shown in Table 6, by first comparing the use of CT labels, we found that the introduction of CT labels significantly improved the model's segmentation performance, especially for smaller organs. Then, by comparing the semi-supervised experiments with and without the use of AMOS unlabeled data, we found that the introduction of AMOS unlabeled data in Stage 2 had little impact on improving the model. Therefore, we chose not to introduce the AMOS unlabeled data too early in Stage 2. Additionally, by incorporating our proposed T1W-to-Multi-sequence label transfer module, the model's DSC and NSD improved by 1% and 1.09%, respectively, demonstrating the effectiveness of this module. We also observed that compared to directly using CT labels and MR pseudo-labels from Stage 2 for supervised learning, first filtering erroneous pseudo-labels using the Anatomy-aware module and then performing semi-supervised learning more effectively improved performance (with DSC improvement of 0.09% vs 0.63% compared to Stage 2).

Finally, we experimented with the number of iterations, as shown in Figure 6. We observed that as the number of iterations increased, the segmentation performance of the model improved, peaking after three iterations, with the most significant improvement observed in the gallbladder, where the DSC increased by 2.92% compared to the first iteration. Beyond three iterations, the performance gains were minimal or even slightly decreased; therefore, we selected the model with three iterations as the final model.

### 4.2   Qualitative results on validation set

As shown in Figure 7, in the cases with good segmentation results, such as Case amos_7261 and Case amos_8141, the baseline method made errors in segmenting the stomach, gallbladder, duodenum, and pancreas. In contrast, our method significantly reduced the segmentation errors in the stomach and gallbladder and improved the segmentation results for the duodenum and pancreas. On the other hand, in cases with poor segmentation results, such as Case amos_8178 and Case amos_0514, the baseline method made severe prediction errors due to the lack of MR data. Although our method also encountered prediction errors, the occurrence of severe prediction errors was greatly reduced compared

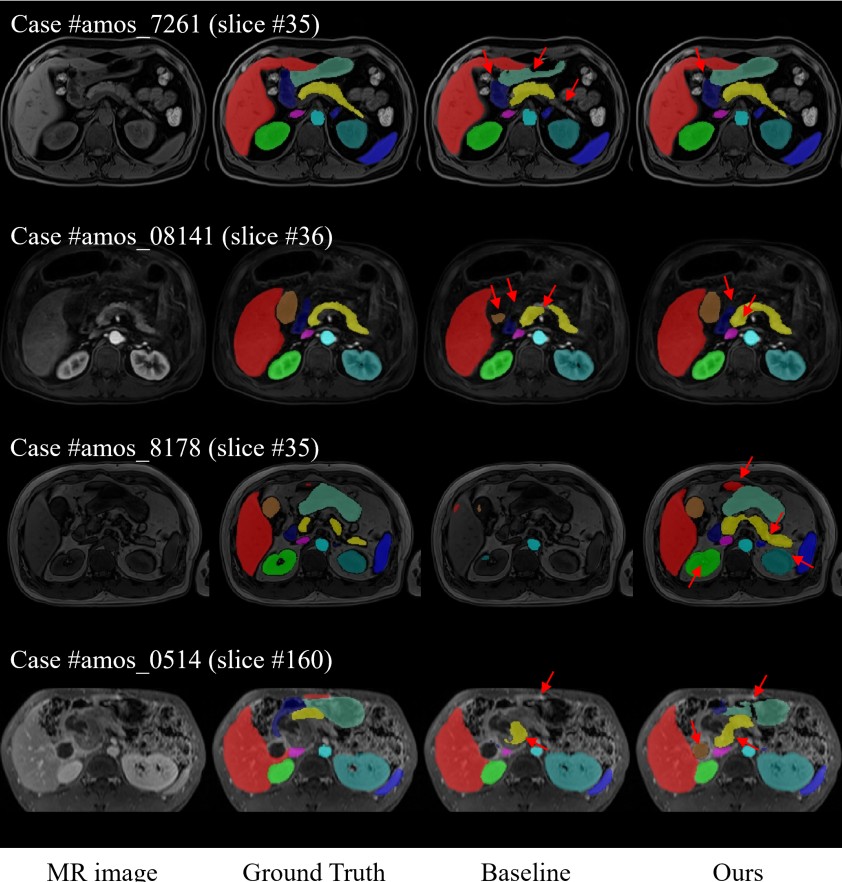

**Fig. 7.** Segmentation result examples: The first and second rows show cases with good segmentation results, while the third and fourth rows show cases with bad segmentation results. The red arrows indicate the areas where segmentation errors occurred.

to the baseline method. However, as shown in Case amos_0514, when the predicted samples have blurred boundaries and complex, disordered structures, our method still encounters prediction errors.

### 4.3 Segmentation efficiency results on validation set

We have submitted our Docker container encapsulating our model to the official challenge. Since our final Docker container submission did not undergo public validation efficiency testing, we conducted local tests on 110 cases. Efficiency metrics for 8 of these cases are shown in the Table 7. The average efficiency metrics are shown in the Table 8.

**Table 7.** Quantitative evaluation of segmentation efficiency in terms of the running them and GPU memory consumption. Total GPU denotes the area under GPU Memory-Time curve. Evaluation GPU platform: NVIDIA GeForce RTX 4090 (24G).

| Case ID | Image Size | Running Time (s) | Max GPU (MB) | Total GPU (MB) |
|---|---|---|---|---|
| amos_0540 | (192, 192, 100) | 9.11 | 3534 | 20018 |
| amos_7324 | (256, 256, 80) | 9.45 | 3540 | 20379 |
| amos_0507 | (320, 290, 72) | 9.1 | 3522 | 19493 |
| amos_7236 | (400, 400, 115) | 10.61 | 3526 | 22968 |
| amos_7799 | (432, 432, 40) | 13.33 | 3575 | 33938 |
| amos_0557 | (512, 152, 512) | 15.51 | 3563 | 35943 |
| amos_0546 | (576, 468, 72) | 10.44 | 3534 | 22395 |
| amos_8082 | (1024, 1024, 82) | 22.36 | 3539 | 50420 |

**Table 8.** Efficiency evaluation results of our submitted docker. All metrics reported are the average values on 110 validation cases.

| Time | GPU Memory | AUC GPU Time | CPU Utilization | AUC CPU Time | RAM | AUC RAM Time |
|---|---|---|---|---|---|---|
| 11.8 | 3536.7 | 26888.4 | 96.89 | 262.62 | 19673.5 | 218337.4 |

### 4.4 Results on final testing set

The final testing results for the proposed method in the FLARE 2024 challenge are summarized in Table 9. The table presents the performance metrics of the method, including the Dice Similarity Coefficient (DSC), Normalized Surface Distance (NSD), inference time, and GPU memory usage. Each metric is reported with both the mean and standard deviation (Mean ± Std), as well as the median along with the first and third quartiles (Median (Q1, Q3)).

**Table 9.** Final testing results of the proposed method on the FLARE 2024 challenge.

| Metric | Mean ± Std | Median (Q1, Q3) |
|---|---|---|
| DSC (%) | 73.3 ± 12.2 | 76.4 (68.3, 82.5) |
| NSD (%) | 77.3 ± 14.6 | 81.0 (71.2, 88.4) |
| Inference Time (s) | 13.2 ± 4.7 | 12.0 (10.3, 14.5) |
| GPU Memory (MB) | 803958.4 ± 321077.0 | 716623.5 (617594.9, 892799.6) |

### 4.5   Limitation and future work

Despite our model achieving relatively satisfactory segmentation performance initially, there are still several shortcomings and areas for improvement in our research, as outlined below.

**Potential for Improvement in Style Transfer.** Our focus has been on leveraging anatomical knowledge and dataset characteristics to address the problem of unsupervised domain adaptation for multi-sequence MRI multi-organ segmentation. In terms of the structure of the style transfer network, we only used the basic CycleGAN framework and did not incorporate the latest style transfer frameworks, such as diffusion models [39], which might have potentially improved our results. Additionally, due to time constraints, we only attempted style transfer for T1W, which has poor contrast. In the future, we will explore using more advanced style transfer networks and attempt CT-to-other-sequence transfers to enhance model performance.

**Underutilization of CT Pseudo-Labels.** It is worth noting that we only used manually annotated labels and a portion of pseudo-labels containing more body information (slice count > 150). For the remaining pseudo-labels, considering that our goal is to segment MR sequences, we did not introduce a large number of CT pseudo-labels. As a result, a substantial amount of high-quality CT pseudo-labels was left unused. In future work, we will further explore how to effectively utilize CT pseudo-labels to assist in the training of MR sequence segmentation models.

**Optimization of Method Steps.** Although the algorithms and models we used are simple, our method involves multiple steps and requires training multiple models, resulting in lengthy training times. In the future, we can optimize some steps, such as combining the style transfer and segmentation tasks in Step 1, to reduce unnecessary training time expenditure.

## 5   Conclusion

We combined unsupervised domain adaptation, registration, semi-supervised learning, and abdominal anatomical structure constraints to propose a simple

three-stage multi-sequence MR segmentation method. First, our method achieves partial label sharing among multi-sequences MR through registration and alignment detection of kidney labels. Next, we employ semi-supervised learning to fully utilize unlabeled data. Finally, we use an anatomical constraint filtering module and iterative training to refine pseudo-labels and further improve model performance. Our method was validated on the large-scale annotated dataset from the MICCAI FLARE 2024 challenge, achieving good segmentation performance in the abdominal organ segmentation task.

**Acknowledgements** The authors of this paper declare that the segmentation method they implemented for participation in the FLARE 2024 challenge has not used any pre-trained models nor additional datasets other than those provided by the organizers. The proposed solution is fully automatic without any manual intervention. We thank all data owners for making the CT scans publicly available and CodaLab [40] for hosting the challenge platform. This work was funded by the Key R&D Program of Guangdong Province (No.2021B0101420006); National Natural Science Foundation of China (No.82472051); National Natural Science Foundation for Young Scientists of China (No.82102019).

## Disclosure of Interests

The authors declare no competing interests.

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

**Table 10.** Checklist Table. Please fill out this checklist table in the answer column.

| Requirements | Answer |
| --- | --- |
| A meaningful title | Yes |
| The number of authors ($\leq 6$) | 5 |
| Author affiliations and ORCID | Yes |
| Corresponding author email is presented | Yes |
| Validation scores are presented in the abstract | Yes |
| Introduction includes at least three parts: background, related work, and motivation | Yes |
| A pipeline/network figure is provided | Figure 2 |
| Pre-processing | Page 4 |
| Strategies to use the partial label | |
| Strategies to use the unlabeled images. | Page 6 – 10 |
| Strategies to improve model inference | Page 10 |
| Post-processing | Page 10 |
| The dataset and evaluation metric section are presented | Page 10 |
| Environment setting table is provided | Table 1 |
| Training protocol table is provided | Table 2, 3 |
| Ablation study | Page 11 – 15 |
| Efficiency evaluation results are provided | Table 7, 8 |
| Visualized segmentation example is provided | Page 7 |
| Limitation and future work are presented | Yes |
| Reference format is consistent. | Yes |