# OpenReview forum: "Joint Unsupervised Domain Adaptation and Semi-Supervised Learning for Multi-Sequence MR Abdominal Organ Segmentation"
_MICCAI.org/2024/Challenge/FLARE — FLARE 2024 withMinorRevisions_

### Official Review · Reviewer_HMUK · 2025-01-24
**Comments**

**Rating:** 8
**Confidence:** 5

**Review:**

The authors present a novel framework for unsupervised segmentation that integrates three key components: (1) CT-to-T1W image translation coupled with pseudo-label generation, (2) label transfer and multi-sequence segmentation, and (3) anatomy-aware refinement through iterative learning. By leveraging these advanced technologies, the method achieves superior performance and offers an effective solution for unsupervised domain adaptation (UDA) tasks.

However, several aspects of the framework require further elucidation:

1.What are the quantitative impacts of the registration process? Given that the AMOS dataset lacks modality information for volumes, how does the framework address this challenge?

2.In Table 5, does the baseline model using only CT images imply the use of generated synthetic MR images?

3.What is the scope or context in which the 498 CT volumes were applied?

---

> ### Author Response · Authors · 2025-03-26
> **Response to Reviewer HMUK**
>
> We sincerely thank Reviewer HMUK for the valuable comments and constructive feedback. Below are our point-by-point responses addressing the concerns raised.
>
> **Comment 1: What are the quantitative impacts of the registration process? Given that the AMOS dataset lacks modality information for volumes, how does the framework address this challenge?**
>
>    Thank you for your insightful comment and for raising the important question about the quantitative impact of the registration process, as well as how the framework addresses the challenge posed by the lack of modality information in the AMOS dataset.
> We believe that the ablation study presented in Table 6, particularly the comparison between Semi(w/o AMOS) and T+Semi(w/o AMOS), sufficiently addresses this issue. The T1W-to-Multi-sequence label transfer module consists of both a registration process and a kidney-aligned filtering algorithm. The assumption behind this module is that the modalities in the dataset are misaligned and require registration for proper alignment. In extreme cases, where all modalities are misaligned, whether or not the registration algorithm is used is equivalent to whether or not to use the label transfer module. In such cases, samples that are not registered would not pass through the kidney-based filtering algorithm, resulting in a degradation to Semi(w/o AMOS). Furthermore, we observed that by combining both components, our method showed a 1% improvement, which implies that without registration, the improvement would be less than 1%.
> Regarding the AMOS dataset, due to the lack of modality information, we only introduce it in the final iterative learning stage, excluding it from the label transfer module. In Table 6, Semi and Semi(w/o AMOS) correspond to the ablation experiment for whether to include AMOS in stage 2, while Supervised and A+Semi(iter=1) represent ablation experiments for how AMOS is used in stage 3 (by either using pseudo-labels directly from stage 2 for supervised learning or filtering pseudo-labels with the Anatomy-aware module before semi-supervised learning).
>
> **Comment 2:In Table 5, does the baseline model using only CT images imply the use of generated synthetic MR images?**
>
> We apologize for the confusion regarding whether the baseline model used synthetic MR images generated from CT data. In fact, the baseline model was trained solely using the original CT images and their corresponding labels. To clarify this point, we have revised the manuscript as follows:"We found that when the model was trained solely on CT labels and the original CT images (window level: 40, window width: 400, scaled to a grayscale range of 0 to 255), without any involvement of MR datasets or synthetic MR images, it still demonstrated a certain ability to segment MR images."
>
> **Comment 3:What is the scope or context in which the 498 CT volumes were applied?**
>
> Thank you for pointing out the need for clarification.We have provided additional clarification in Figure 2 and Section 2.3, specifying that these samples were only used in tage 1: "As shown in Figure 2, style transfer is conducted exclusively from CT to the T1W. We selected 100 manually annotated CT samples and 398 CT pseudo-labeled samples (sorted by name) as the source domain for style transfer (Note that the 498 CT samples, which have been processed through Translation registration, were used exclusively in the first stage.)"
>
> We hope these clarifications adequately address your concerns. Once again, we greatly appreciate your insightful comments, which have contributed to strengthening our work.

---

### Official Review · Reviewer_yAgk · 2025-01-25
**Review of "Joint Unsupervised Domain Adaptation and Semi-Supervised Learning for Multi-Sequence MR Abdominal Organ Segmentation"**

**Rating:** 8
**Confidence:** 5

**Review:**

The authors propose a three-stage multi-sequence MR abdominal organ segmentation method:
（1）Performing style transfer from CT to T1W.
（2）Introducing a T1W-to-Multi-sequence label transfer module, which determines whether to share labels based on kidney similarity.
（3）Using CT and MRI data for semi-supervised learning.

However, there are still some weaknesses in this paper:

（1）There are invalid superscripts in Tables 2 and 3.

（2）The authors said "To improve the model’s learning of features from other regions of the body and enhance its robustness, we introduce manually annotated samples and pseudo-labeled samples with over 150 slices from the CT dataset."

The paper lacks ablation studies on the absence of CT labels for semi-supervised learning.

---

> ### Author Response · Authors · 2025-03-26
> **Response to Reviewer yAgk**
>
> Thank you for your valuable review comments. Below are our responses to the issues you raised:
>
> **Comment 1: There are invalid superscripts in Tables 2 and 3.**
>
> Thank you for pointing out this issue. Regarding the invalid superscripts in Tables 2 and 3, we have carefully reviewed and made the necessary corrections to ensure that all superscripts are valid, accurate, and consistent throughout the tables:
>
>
> **Comment 2: The authors said "To improve the model’s learning of features from other regions of the body and enhance its robustness, we introduce manually annotated samples and pseudo-labeled samples with over 150 slices from the CT dataset."**
>
> Thank you for your thoughtful and constructive comment. Regarding the point you raised about the statement, "To improve the model’s learning of features from other regions of the body and enhance its robustness, we introduce manually annotated samples and pseudo-labeled samples with over 150 slices from the CT dataset," and your concern about the lack of ablation studies on the absence of CT labels for semi-supervised learning, we have carefully addressed this issue. Specifically, we have added the relevant ablation experiments in Table 6 and provided a detailed explanation in Section 4.1, clarifying the impact of excluding CT labels in the semi-supervised learning phase:
> “As shown in Table 6, by first comparing the use of CT labels, we found that the introduction of CT labels significantly improved the model’s segmentation performance, especially for smaller organs.”
>
> We sincerely appreciate your careful review, and your feedback has been invaluable in improving the quality of the paper. We will continue to focus on the details to ensure the paper is more rigorous and complete.

---

### Official Review · Reviewer_zN7G · 2025-03-02
**Review of "Joint Unsupervised Domain Adaptation and Semi-Supervised Learning for Multi-Sequence MR Abdominal Organ Segmentation“**

**Rating:** 8
**Confidence:** 5

**Review:**

This study proposes a three-stage method for multi-sequence MR abdominal organ segmentation, incorporating unsupervised domain adaptation, semisupervised learning, registration, and anatomical structure constraints.
Well writen article !
1. Fig3's title is a bit informal.

---

> ### Author Response · Authors · 2025-03-26
> **Response to Reviewer zN7G**
>
> **Comment 1:Fig3's title is a bit informal.**
>
> Thank you for your thoughtful and detailed feedback on the title of Fig. 3. We appreciate your suggestions for improving the formality of the caption. As per your recommendation, we have revised the title as follows:
> “Figure 3: Structure and Layer Configuration of the 3D U-Net Model Employed in the Study”
> This adjustment aims to make the caption more formal and in line with academic standards. Thank you again for your valuable input.

---

### Official Review · Reviewer_Pf9T · 2025-03-04
**Review of "Joint Unsupervised Domain Adaptation and Semi-Supervised Learning for Multi-Sequence MR Abdominal Organ Segmentation“**

**Rating:** 8
**Confidence:** 5

**Review:**

The paper proposes a three - stage method integrating unsupervised domain adaptation, semi - supervised learning, and anatomical constraints for multi - sequence MR abdominal organ segmentation. It enables label sharing among MR sequences, effectively utilizes unlabeled data, and improves pseudo - label accuracy. The comments are listed below:

(1) In the introduction, when discussing existing methods, the tenses shift. "Yang et al. [9] proposed an unsupervised domain adaptation technique... Similarly, Xu et al. [10] employed content and style separation... " Here, the past tense "proposed" and "employed" is used. But later, "However, these methods have rarely fully leveraged the characteristic of different sequences... " uses the present perfect tense.
(2) In the sentence "Our method achieved an average score of 81.60% and 89.83% for the organ DSC and NSD on the validation set and the average running time and area under GPU memory - time curve are 11.80s and 26888MB, respectively.", the switch from "achieved" (past tense) to "are" (present tense) is a bit inconsistent.
(3) In Table 2, row "Optimizer", one of the beta_1 should be beta_2 as the Adam optimizer typically has two different momentum parameters.

---

> ### Author Response · Authors · 2025-03-26
> **Response to Reviewer Pf9T**
>
> **Comment 1:In the introduction, when discussing existing methods, the tenses shift. "Yang et al. [9] proposed an unsupervised domain adaptation technique... Similarly, Xu et al. [10] employed content and style separation... " Here, the past tense "proposed" and "employed" is used. But later, "However, these methods have rarely fully leveraged the characteristic of different sequences... " uses the present perfect tense.**
>
> Thank you very much for your careful review and for pointing out the issue with the inconsistent use of tenses in the introduction. We truly appreciate your attention to detail. Upon revisiting the manuscript, we have recognized the inconsistency in the tense usage and have made the necessary correction. Specifically, we have revised the sentence as follows:"However, these methods rarely fully leveraged the characteristic of different sequences of the same sample with highly similar structures within multi-sequence MR datasets."We hope this revision resolves the issue, and we are grateful for your valuable suggestion.
>
> **Comment 2: In the sentence "Our method achieved an average score of 81.60% and 89.83% for the organ DSC and NSD on the validation set and the average running time and area under GPU memory - time curve are 11.80s and 26888MB, respectively.", the switch from "achieved" (past tense) to "are" (present tense) is a bit inconsistent.**
>
> Thank you very much for your careful review and for pointing out the inconsistency in tense usage. We sincerely apologize for this oversight. Upon reviewing the manuscript, we have made the necessary correction as per your suggestion. Specifically, we have revised the sentence to:
> "Our method achieved an average score of 81.60% and 89.83% for the organ DSC and NSD on the validation set, and the average running time and area under GPU memory-time curve were 11.80s and 26888MB, respectively."
>
> **Comment 3: In Table 2, row "Optimizer", one of the beta_1 should be beta_2 as the Adam optimizer typically has two different momentum parameters.**
>
> Thank you very much for pointing out the issue with the momentum parameter in the "Optimizer" row of Table 2. We sincerely appreciate your attention to this important detail. Upon review, we realized that one of the values was mistakenly listed when it should have been, as the Adam optimizer indeed uses two different momentum parameters.We have corrected this error in the manuscript, and the revised table now reflects the accurate information. We apologize for this oversight and are grateful for your careful review.

---

### Decision · Program_Chairs · 2025-03-20

**Decision:**

Accept

**Comment:**

Please carefully address the reviewers' comments in the revision.

---

> ### Author Response · Authors · 2025-03-29
> **Response to Program Chairs**
>
> Thank you for your feedback. We have carefully addressed all the reviewers' comments and revised the manuscript accordingly. A detailed response to each point is included in the revised submission, and the requested test results have been incorporated into the manuscript.